# Remarkable Recycling Process of ZnO Quantum Dots for Photodegradation of Reactive Yellow Dye and Solar Photocatalytic Treatment Process of Industrial Wastewater

**DOI:** 10.3390/nano12152642

**Published:** 2022-07-31

**Authors:** Walied Mohamed, Hala Abd El-Gawad, Hala Handal, Hoda Galal, Hanan Mousa, Badr El-Sayed, Saleh Mekkey, Ibrahem Ibrahem, Ammar Labib

**Affiliations:** 1Photochemistry and Nanomaterials Lab, Inorganic Chemistry Department, National Research Centre, Cairo 12622, Egypt; hally.handal@gmail.com (H.H.); hrgalal@hotmail.com (H.G.); hmousa3@hotmail.co.uk (H.M.); ammar_al@yahoo.com (A.L.); 2Department of Chemistry, Faculty of Science and Arts, King Khalid University, Mohail Assir, Abha 61421, Saudi Arabia; habduljwaad@kku.edu.sa; 3Inorganic Chemistry, Chemistry Department, Faculty of Science Al-Azhar University, Cairo 11884, Egypt; badrelsayed@gmail.com (B.E.-S.); salehsdmm@yahoo.com (S.M.); iali@bu.edu.sa (I.I.); 4Chemistry Department, Faculty of Science and Arts in Al-Mandaq, Al-Baha University, Al-Baha 1988, Saudi Arabia

**Keywords:** industrial wastewater, solar photocatalytic process, recycling process, zinc oxide quantum dots, reactive yellow dye

## Abstract

The mineralization of five industrial sunlight-exposed wastewater samples was investigated, and the recycling process of ZnO quantum dots (ZQDs) for five reusable times was estimated under the approved Egyptian Environmental Law COD (Chemical Oxygen Demand), which has to be less than 1000 ppm. An improved sol-gel process at a low calcination temperature that ranged between 350 and 450 °C was employed to synthesize ZnO quantum dots (ZQDs). The purity, high crystallinity, and structure of the prepared catalysts were determined by TEM and XRD analysis. The energy bandgap, the crystal size values, and the surface area for Z1 and Z2 were determined based on the TEMs, DRSs, and EBTs, which were equal to 6.9 nm, 3.49 eV, and 160.95 m^2^/g for Z1 and 8.3 nm, 3.44 eV, and 122.15 m^2^/g for Z2. The investigation of the prepared samples was carried out by studying the photocatalytic activity and photoluminescence, and it was found that the degradation rate of reactive yellow dye as an industrial pollutant of the Z1 sample was significantly higher than other samples, by 20%. The data collection has shown that photocatalytic efficiency decreases with an increase in the crystallite size of ZQDs.

## 1. Introduction

The global environmental management industry still has a disability in the face of massive quantities of emissions from different manufacturing factories. One of the worst wastewater materials is industrial wastewater from the dye industry, which has an unbiodegradable impact on the environment and human beings. Reactive yellow dye is a type of color used in most textile and printing sectors. It is also known as a coloring model to be used to examine photocatalytic efficiency for photocatalysts [1,2,3].

The photocatalytic activity of the photodegradation processes of organic pollutants depended on using metal oxides as basic materials regarded as distinguishable impacts in wastewater management. Nano-heterogeneous photocatalysis exhibits a wide-bandgap, environmentally friendly, and relatively inexpensive biocompatibility of nanostructured semiconducting oxides, which are functionally ascribed to the method. Additionally, these compounds can break down different types of pollutants under Ultra-Violet light. As a nano-heterogeneous catalyst, nano zinc oxide with a wide band gap (3.37 eV) at room temperature is considered the most efficient photocatalyst for water splitting, wastewater treatment, and photodegradation of organic dyes [4,5,6,7,8,9,10,11].

To create high-quality ZnO quantum dots, researchers have studied a number of different preparation processes. These methods are physical vapor deposition [12,13,14], which comprises thermal evaporation, molecular beam epitaxy, sputtering, and laser ablation [15,16,17]; chemical vapor deposition (CVD), which comprises metal-organic CVD, mist CVD, hot filament CVD, and atomic layer deposition [12,18,19]; and chemical solution synthesis, which comprises sol-gel synthesis, hydrothermal growth, and spray pyrolysis [2,20,21,22,23]. In particular, the parameters of these synthesis procedures are tuned in order to create ZnO quantum dots with the required physical and chemical properties, as well as excellent stability and compatibility. In contrast, parameter optimization often involves a trade-off between the desired features.

Physical, chemical, and photoluminescent properties of Zinc Oxide quantum dots (ZQDs) are sometimes not observed because of their 10 nm size, which is small for micro-molecules photocatalysts. Their 3-D structure is extremely important for the fabrication of ZQDs, resulting in enhanced optoelectronic properties such as radiative lifetime and exciton energy, which often depend on the quantum impact. Furthermore, the quantitative theory “particles in a box” explain that particle size changes are due to changes in the electronic structure, where holes and electrons are spatially confined, and energy is discrete, so light emission and absorption are the result of reduced particle size and are significantly enhanced [24,25,26]. Further, ZQDs can be used as an advanced technique for cancer therapy, as in bio-science cell tagging, the antibacterial agent and anti-fouling, and ZQDs [27,28,29,30,31]. Recent work has shown that ZQDs can be used in QD TV (4 and 8 k TVs) as lighted diodes, photodetectors, high-efficiency solar cells (with 60 percent expected efficiency), UV laser detectors, and in some new applications, as well as thin-film transistors [32,33,34]. The surface, structure, and visual properties of ZQDS-prepared samples are studied in this current work. A recent evaluation phase for measuring the photocatalytic activity and tuning the emission of the ZQDS samples was observed and measured via the photodegradation process of the reactive yellow dye by the Xenon photoreactor.

In the present study, the surface composition and optical properties of the prepared ZQDS were analyzed. Further steps toward the assessment and calculation of the photocatalytic activities of the prepared samples are observed by studying and quantifying the photodegradation process of the reactive yellow color dye under Xenon irradiation. Further, the photocatalytic efficiency of the ZQDS samples in industrial wastewater treatment at dyeing companies was estimated under direct sunlight, like a solar photocatalytic process.

## 2. Materials and Methods

### 2.1. Materials

All the chemicals used during this work have been analytically classified by Sigma-Aldric (St. Louis, MO 68178, United States) and Alfa Aesar (Erlenbachweg 276870, Kandel, Germany). They have not been further purified, such as Zinc acetate dihydrate, lithium hydroxide monohydrate, acetic acid, ammonia, propanol, n-heptane, and ethanol HPLC.

### 2.2. Preparation of Zinc Oxide Quantum Dots Samples

This synthesis method of ZQDs depended on using zinc II acetate dihydrate, lithium hydroxide-monohydrate, and ethanol. First, 2 moles of zinc acetate dihydrate are dissolved in 100 mL of ethanol in an ultrasonic bath at 60 °C for 90 min, representing solution (A). In the same volume of ethanol, 1.15 g of lithium hydroxy is dissolved and described as solution (B). Then, solution (B) was slowly added to solution (A) under vigorous stirring (9000 rpm) at atmospheric pressure and room temperature to form solution (C). When n-heptane was added to the solution (C), a white precipitate appeared under vigorous stirring, and the precipitate was isolated, extracted, and then dispersed into ethanol, which was then recentrifuged to complete the elimination of excess lithium ions. The ZQDs are repeatedly washed away by n-heptane and ethanol to eliminate the remaining acetate ions. A clear white precipitate was obtained and dried at 26 °C (room temperature), then calcinated in an oven for 3 h at 350 and 450 °C. Finally, a purified ZQDs (Z1 and Z2) powder was ready for potential measurements.

### 2.3. Characterization

A PANalytical X-ray diffractometer (X’pert Pro MPD, Los Altos, CA 94022, USA) with a Cu-Kα target (Cu Kα = 0.154 nm, 40 mA, 50 kV, recorded in 0.017° steps, 100 s/step) measured the crystallinity properties of the ZQDs samples. The following Debye–Scherrer formula for measuring the crystallite size of ZQDS is applied: (1)$$D=\frac{0.9\lambda }{\beta \cos\theta }$$
where 2*θ* is the diffraction angle at the position of the maximum peak, *λ* is the radiation wavelength used in Cu Kα (0.15406 nm), *β* is the full width at the half-maximum of the peak (FWHM), and K (=0.9) is Scherrer’s constant [2,35]. A high-resolution transmission electron microscope HRTEM, Philips/FEI BioTwin CM120, was used to study the particle size and shape of ZQDs. In addition, a JASCO V-570 Rel-00 USA (Easton, MD 2160, USA) spectrophotometer was used to measure photophysical observations such as the fluorescence rate constant (kf) of the prepared sample and the kinetic fluorescence intensity slope ratio to illumination time was calculated; the sample was measured using a Shimadzu UV260 spectrometer (Columbia, MD 21046, USA). A Micrometrics Tristar 3000 device, Micromeritics Instrument Corporation (Norcross, GA 30093, USA), was used for BET-specific surface area. The prepared samples were analyzed in a nitrogen atmosphere (adsorption-desorption isotherms 77 K), where the drying process was carried out for 1 h at 150 °C before the evaluation of the eliminated moisture levels in the prepared samples. In the Barrett, Joyner, and Halenda methods, the average pore size and pore size distribution were calculated, and the specific surface area was determined by the following equation:(2)$$S=\frac{6}{d\times \rho }$$
where ρ is the density of ZnO (5.63 g/Cm^3^), d is the average diameter of particles, and S is the BET-specific surface area [36].

Further, the expression of Jan Tauc was used to measure the bandgap energy values for the catalysts prepared by the following equation:(3)$${\left(\alpha h\nu \right)}^{2}=A\left(h\nu -E_{g}\right)$$
where *A* is a constant, *h* is the Plank constant, *E_g_* is the bandgap, and *α* is the absorption coefficient.

A Photoreactor X 100 with a xenon arc lamp produced by Engineering Egyptian Co. with a water-cooling system is used to prevent the temperature influence of the lamp during all photodegradation processes where the intensity of the xenon lamp was 70 W/Cm^2^, and its wavelength ranged from 200 to 1100 nm.

A multi-parameter bench photometer evaluated the mineralization of reactive yellow dye and real industrial wastewater samples (from one of the factories, Cairo, Egypt) in the presence of ZQDs samples. Multiparameter Bench Photometers (COD series 99), Hanna (Allendale, SC 29810, USA), determined the chemical oxygen demand (COD).

### 2.4. Photocatalytic Activity Performance

The photocatalytic performance was tested as a model of industrial dye by the photodegradation of reactive yellow dye: 0.25 g ZQDs was distributed into 250 mL of the investigated dye solution (4 × 10^−5^ M; pH = 6.5) with stirring in the dark for 20 min to reveal that the absorbability of the dye on the catalyst surface was negligible, as the decreasing percent does not exceed 0.8%, and the photodegrading process was carried out at 70 Watts. After 20 min of the irradiation phase, the ZQDs particles were strongly separated from the suspension solution by direct centrifugation at 8000 rpm. The photodegradation process rates of reactive yellow dye in the presence of ZQDs were spectrophotometrically detected as the following equation
(4)$$\ln\;\left(C_{0}/C\right)=k_{app}t$$
where *C* is the dye concentration, *C*_0_ is the initial dye concentration, *t* is time, and *k_app_* is the apparent rate constant for the photodegradation process.

Further, the fluorescence rate of ZQDs was improved by the photooxidation process that transforms correct quantities of coumarin dye to 7-hydroxy coumarin as follows: 0.5 g/L of ZQDs was immersed in a coumarin dye solution (1.0 × 10^−4^ M) and irradiated by a UV reactor (80 Watt) with stirring at 9000 rpm. Then, the fluorescence spectrum of the solution (*λ*_ex_ = 332 nm) was measured over determined periods after illumination. The coumarin samples have been completely inactive after the course of UV irradiation and are viewed as a blank sample in the absence of ZQDs particles that represent photocatalysts [24].

### 2.5. Solar Photocatalytic Activity

The solar photocatalytic efficiency of the real industrial wastewater was measured at pH = 7.1 from a textile factory, and sunlight with a UV radiation of 3.1 W/Cm^2^ was used, while the visible light was 1198 lux. The efficiency of solar mineralization, measured by a highly active sample with time t, was evaluated using the following equation:(5)$$\mathrm{Solar}\;\mathrm{mineralization}\;\mathrm{efficiency}=100\;\times \;\left[{\left(COD\right)}_{0}-{\left(COD\right)}_{t}\right]/{\left(COD\right)}_{0}$$
where chemical oxygen demand, or *COD*, is the ability of the water to consume oxygen during the breakdown of organic matter. In other words, the amount of oxygen necessary to oxidize the organic stuff in a given volume of water (*COD*); *COD_t_* and *COD*_0_ are the *COD* value at time *t* and initial at 0, respectively [7].

## 3. Results and Discussion

### 3.1. XRD

As illustrated in Figure 1, X-ray diffraction analysis was used to examine the crystal properties of the produced quantum dot samples. The XRD patterns were completely matched to the wurzite structure of ZnO, index card number 36–1451 from JCPDS-ICDD [27,37,38]. The observed diffraction peaks at 2θ were identified as the (100), (002), (101), (102), (110), (103), and (112) planes, respectively. No additional distinctive peaks of impurities or other secondary phases were observed in the patterns, indicating the great purity and crystallinity of the samples produced for the experiment.

The values obtained are described in Figure 1b, which demonstrate that the crystallite size measured increases gradually as the rinsing temperature increases, whereas the average crystallite size was 6.9 and 8.3 nm, respectively, for Z1 and Z2 samples.

### 3.2. TEM

The HRTEM of the prepared samples shown in Figure 2 is based on the morphology and particle size of ZQDs. TEM images demonstrated that ellipsoidal prolonged, ultra-finite nanoparticles with excessive crystallinity had been formed.

For Z1 and Z2, the particle size is 6.9 and 8.3 nm, respectively. The average particle size of the ZQDs samples is significantly greater than that obtained from the XRD data, as the calcination temperature rose from 350 to 450 °C for Z1 and Z2, respectively. This may be due to the aggregation of powder samples in the copper-coated grid [24,39,40].

### 3.3. Bandgap

The absorption spectrum measured in the UV region is shown in Figure 3. The Z2 result was found to have higher absorption than Z1 from a continuum. In comparison, the absorption rates rose by increasing the temperature of the calcination. One of the most important optical property tests is the bandgap value detection, which represents an effective element in the catalytic activities of ZQDs samples prepared by using the Jan Tauc equation.

Bandgap energy values are estimated as 3.49 and 3.44 eV for Z1 and Z2 samples, respectively, which are clearly higher than ZnO bulk (3.37 eV) [2,41]. The correlation between BET, Eg, and ZQDs samples is represented in Figure 4; based on the particle size of the prepared samples, the BET values are directly commensurate with the band gap energy and inversely commensurate with the crystallite sizes of the photocatalyst prepared.

As a result of the surface area increase, the mass transfer accelerated and produced further reaction sites, which will increase the reaction catalytic efficiency for Z1 samples compared to the other sample (Z2).

ZQDs specimens’ sizes of had 160.95 and 120.15 m^2^/g for the Z1 (6.9 nm) and Z2 (8.3 nm) samples, respectively, where these samples were synthesized at various calcination temperatures of 350 and 450 °C.

### 3.4. Photocatalytic Activity by Spectrophotometrically Method

The photodegradation of reactive yellow dye test is performed in the presence of ZQDs by a xenon photoreactor and spectrophotometrically detected as in Appendix A.

Figure 5 shows a linear relationship between the irradiation time and ln (*C*_0_/*C*). The first-order kinetics follow photocatalytic studies with reactive yellow dye and the correlation between maximum photodegradation percentage, photodegradation times, and photodegradation rates of the prepared ZQDs samples are represented in Figure 6. The findings confirmed that photodegradation process rates increases as ZQD’s bandgap values increase and the ZQDs particle size decreases (Table 1 and Figure 4). The photodegradation rate for Z1 is 17 percent higher than Z2. The findings thus verified BET values and the crystallite size suggesting that the surface area average increases with decreases in the crystallite size due to increases in the photocatalytic activity of ZQDs samples.

The highest photodegradation process percentage of reactive yellow dye was 93% and 88.5% in the presence of samples prepared from ZQDs (6.9 and 8.3 nm), respectively, as the rates of photodegradation decreased with the increasing size of samples prepared from ZQDs. In the presence of the prepared ZQDs samples, the formation of the hydroxyl radicals is the main core of the photodegradation process mechanism of the reactive yellow dye, which was clarified in previous work [2,24].

In the ZQDs samples, the electrons of the conduction bands and the holes of the valance band were formed by radiation light (hν) on their surface, where hydroxyl radical (HO^·^) is created when combined with a positive hole (h^+^) and/or H_2_O/(H_2_O_2_). Further, O_2_ molecules are absorbed onto the photocatalyst surface by the conductive band electrons to create O_2_^−^ (the superoxide anion radical). The (O_2_^−^) and (HO) created to represent all available oxidants for the degradation of reactive yellow dye. These oxidant ions incorporated in the ZnO lattice produce more electrons and gorgeous hole traps, which increase the charge carriers, consequently suppressing the recombination of photoinduced electrons and holes by means of a high enhancement in the photocatalytic activities. Consequently, the recombination phase of electron-hole delays and high photocatalytic activity are detected as the bandgap of the prepared ZQDs samples increase. The acute electrons in the conductive band, creating radical superoxide anions, are decreased by O_2_ molecules absorbed into the photocatalyst, where the evolved (HO) and (O_2_^−^) is determined; the (O_2_^−^) is formed as a powerful oxidant at the same time as the degradation process of the reactive yellow dye [42]. In a perfect photocatalytic process, organic contaminants are converted into carbon dioxide, water, and mineral acids. When photons equal to or exceed the band gap of ZnO’s surface, photocatalysis occurs [43]. During illumination, electrons from the valence band (VB) (e^−^) are photoexcited to the conduction band (CB), leaving a hole (h^+^) behind (Equation (6)). Excited electrons and holes may swiftly recombine. In the second equation, the photogenerated electron-hole pairs migrate to the surface of Zinc Oxide, decrease hydroxyl radicals (^●^OH), and form superoxide radicals (Equations (7) and (8)). The created superoxide radicals also induce the production of additional (^●^OH) radicals [44,45,46] (Equations (9)–(11)), which are capable of effectively destroying the pollutant (Equation (12)).



(6)
$$\mathrm{ZnO}\;+\;h\upsilon \to e_{\mathrm{CB}}^{-}\;+\;h_{\mathrm{VB}}^{+}$$


(7)
hVB+ + H2O→H+ + OH●


(8)
$$e_{\mathrm{CB}}^{-}\;+\;O_{2}\to O_{2}^{-}$$


(9)
$$O_{2}^{-}\;+\;H^{+}\to {\mathrm{HO}}_{2}^{-}$$


(10)
$${\mathrm{HO}}_{2}^{-}\;+\;H^{+}\to H_{2}O_{2}$$


(11)
H2O2+hυ→2OH●


Organic pollutant + ^●^OH → Intermediate + CO_2_ + H_2_O (12)


Due to its broad band gap, ZnO can only degrade pollutants under UV light. UV radiation is just 4–5% of the solar spectrum, whereas visible light is 4% [47,48,49,50,51].

### 3.5. Photocatalytic Activity by Fluorescent Probe Method

Table 1 describes the fluorescence ratio of ZQDs samples provided by the fluorescent test method, where coumarin photooxidation to 7-hydroxycoumarin and the reaction rate constant (kf) are calculated. Coumarin emission spectra provided ZQDs samples under UV lighting at various times, as shown in Figure 7. Further, as the ZQDs sample’s particle size decreases, the apparent rate continuously increases, as shown in Figure 7, and the correlation between photodegradation and fluorescence rates is represented in Figure 8.

The development of hydroxyl radicals depends on the rivalry between surface water oxidation and the electron-hole recombination in the holes or the hydroxyl group [24]. Consequently, higher electron-hole pair separation efficiency is a referee to a higher rate of creation of (HO^·^) that contribute to declines in the ZQDs particle size. As seen in the optical measurements’ parts (UV-Vis/DR absorption and emission spectra), the quantum dots scale plays a significant role in increases in the isolation of electron-hole pairs. The ZQDs sample (Z1), prepared at the lowest calcination temperature of 350 °C (6.9 nm), has the highest formation of hydroxyl radicals and can effectively prevent the recombination of electrons and holes.

The further from the hydroxylic radical formations, the more effective electron-hole pairs are, and the more efficiently they detach. This reaction is based on a kinetic (kf) pseudo-zero order reaction, which describes the rate of formation of hydroxyl radical coumarin as 7-hydroxycoumarin for photooxidation [52].

The highest photodegradation rate found in the Z1 sample (6.9 nm) is definitely due to its higher surface area, which is higher than the Z2 sample (8.3 nm) because of the number of dye molecules adsorbed on Z1 (Z1 = 6.9 nm) is higher than Z2 (Z2 = 8.3 nm). The findings of the photocatalytic operation of the investigated dye performed to contrast the quantum size effect were apparent. For Z1 (6.9 nm), the fluorescence rate was more than 40 percent higher than that found in Z2. These findings show that the surface area of the ZQDs sample increases as its particle size reduces and the fluorescence rate increases.

### 3.6. Solar Photocatalytic of Real Industrial Wastewater (Application)

Concerning the prepared ZQDs samples, the mineralization efficiencies modified the substantial of numerous real industrial wastewater samples (5 different samples) taken from the industrial textiles dye company in the presence of commercial nano zinc oxide and ZQDs samples (Z1 and Z2) with COD values between 6101 and 7020 (ppm). Figure 9 and Appendix A show the results of this analysis after it was performed for 7 h a day (8 am to 3 pm).

A survey considering the impact of particulate size as the key factor influencing this analysis also included the photocatalytic operation conducted on the actual wastewater effluent in a textile factory using commercial nano ZnO (25–30) nm. The permissible COD value for industrial wastewater at the textile dye companies is 1000 ppm, as specified by Egyptian Environmental Law [53]. Figure 9 shows that all COD values are below the COD limits. Except for the first two samples, those COD values of commercial nano ZnO (25–30 nm) had been above the upper COD limit (less than 1000 ppm) (1299 and 1152 ppm).

Future studies will rely on economic catalyst uses for wastewater treatment units of a combination of various zinc oxide forms (nano and quantum dots) than on ZQDs. More green approaches for processing samples for ZQDs and analyzing them in specialized applications would also be more feasible.

### 3.7. Recycling Processes

#### 3.7.1. Recycling ZQDs during Decontamination of Reactive Yellow Dye

The photodegradation process rates decreased due to the accumulation of the photocatalysts observed via the recurrence of using of the photocatalysts, as shown in Figure 10. The use of ZQDs prepared samples was repeated five times during the recycling of the photodegradation process. Other interesting studies have shown that the repetition variation for Z1 (6.9 nm) is 15% and Z2 (8.3 nm) is 20%. These findings demonstrate the efficacy of the photodegradation processes with the quantum point size effect.

#### 3.7.2. Recycling of ZQDs during Real Industrial Wastewater

Table 2 and Figure 11 evaluated the method of repeating the samples prepared with ZQDs. This procedure is conducted for residue water from 3 February until 11 February 2021 for 7 h a day from 8:30 am until 3:30 pm, where its COD value is equal to 7020 ppm, and the assessment process for recycling the real industrial wastewater treatment process using photocatalysts is performed seven times.

The result shows that with Z1 (6.9 nm) and Z2 (8.3 nm), the mineralization performance still makes COD limits after six and five instances of recycling. Although commercial nano ZnO (25–30 nm) still permitted COD after three times instances of the recycling process, and still in a risky condition allowed COD after four instances of the recovery process (991 ppm). It is obvious that, as a result of the replication of the recycling process, the size of photocatalysts increases during the recycling process.

## 4. Conclusions

ZQDs (6.9 and 8.3 nm) were developed at low calcination temperatures of 350 and 450 °C using a modified sol-gel process. All prepared ZQDs samples, demonstrated by XRD and TEM, have high purity in the single phase, while ZQDs samples’ particle size increased from 6.9 nm (Z1) to 8.3 nm (Z2) where Z1 was calcinated at 350 °C and Z2 at 450 °C. The photocatalytic efficiency of the prepared ZQDs against reactive yellow color degradation approaches 93%, and the photographer rate for Z1 exceeds 17%, while the fluorescence rate for Z1 exceeds Z2 by 17% due to the confinement effect of the prepared ZQDs samples. On the other hand, the mineralization productivity of real industrial wastewater from one of the large textile plants in Egypt and the Middle East followed the approved COD limits in Egypt’s Eco-Law specification.

Furthermore, ZQDs increase in their size in the recycling phase as a recurrence (7 times) of the accumulation of photocatalysts contributes to the increase in the above allowable COD values after 6 and 5 recycling processes. Therefore, due to the high surface area, the containment size effect, the photophysical properties, and photocatalytic action, the samples prepared will contribute to future work as distinctive photocatalysts in recent years.

## Figures and Tables

**Figure 1 nanomaterials-12-02642-f001:**
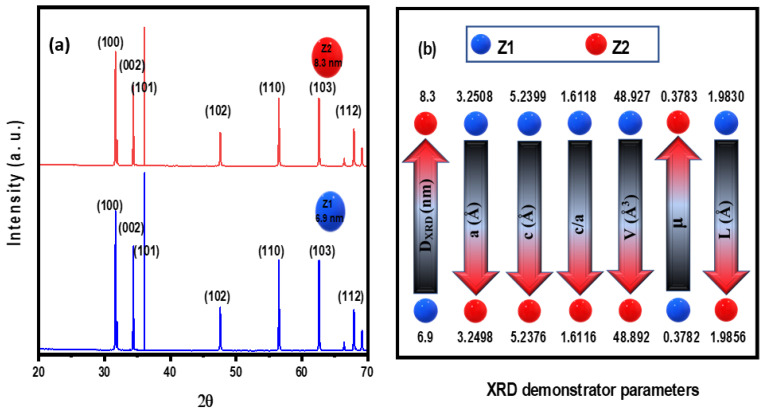
XRD pattern of ZQDs (**a**), XRD demonstrator parameters (**b**).

**Figure 2 nanomaterials-12-02642-f002:**
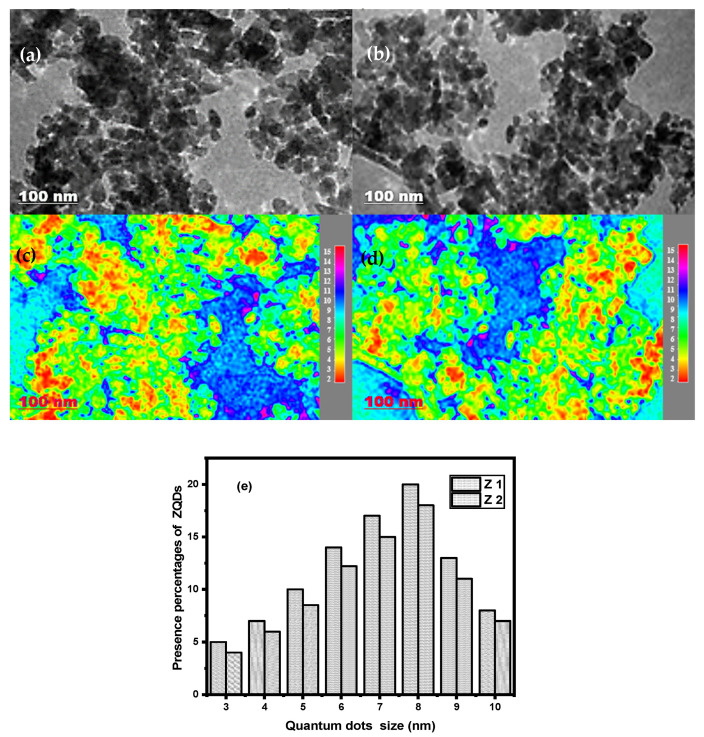
TEM image of different prepared ZQDs samples (**a**,**b**) along with their corresponding particle size distribution map (**c**,**d**) and presence percentages of ZQDs (**e**).

**Figure 3 nanomaterials-12-02642-f003:**
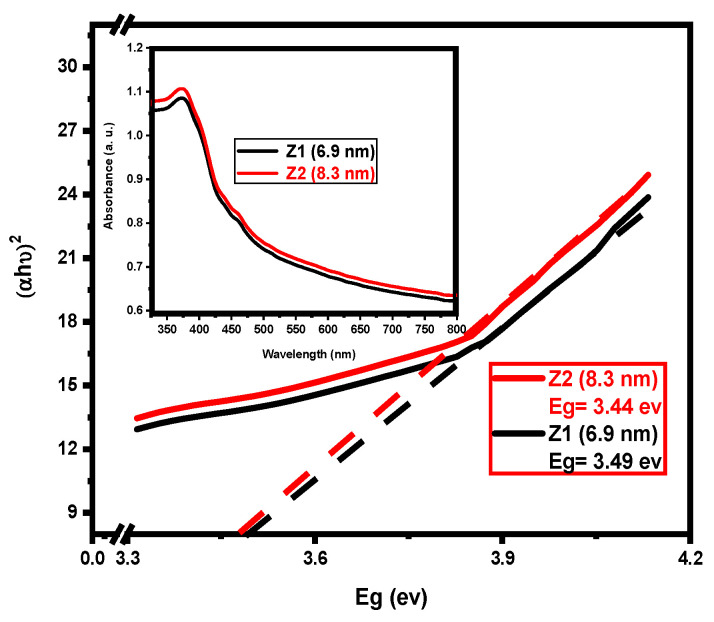
UV-Visible absorption spectra and estimation bandgap energies of ZQDs. The inner image is the absorption spectrum of Z1 and Z2 samples.

**Figure 4 nanomaterials-12-02642-f004:**
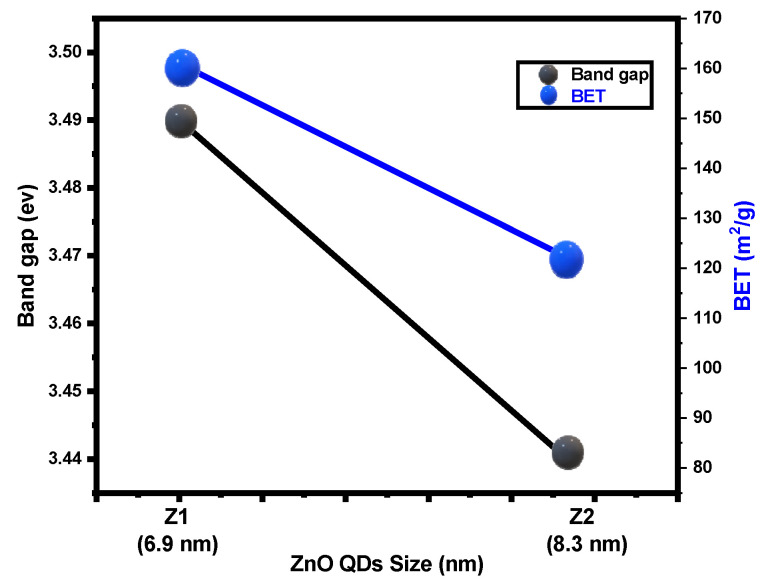
Correlation between BET, Eg, and particle size of ZQDs.

**Figure 5 nanomaterials-12-02642-f005:**
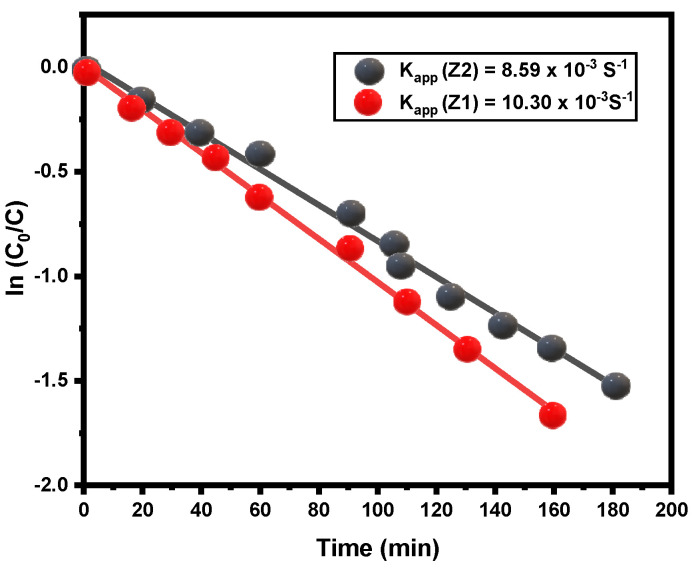
Kinetics plot of the photodegradation rate of reactive yellow dye in the presence of ZQDs.

**Figure 6 nanomaterials-12-02642-f006:**
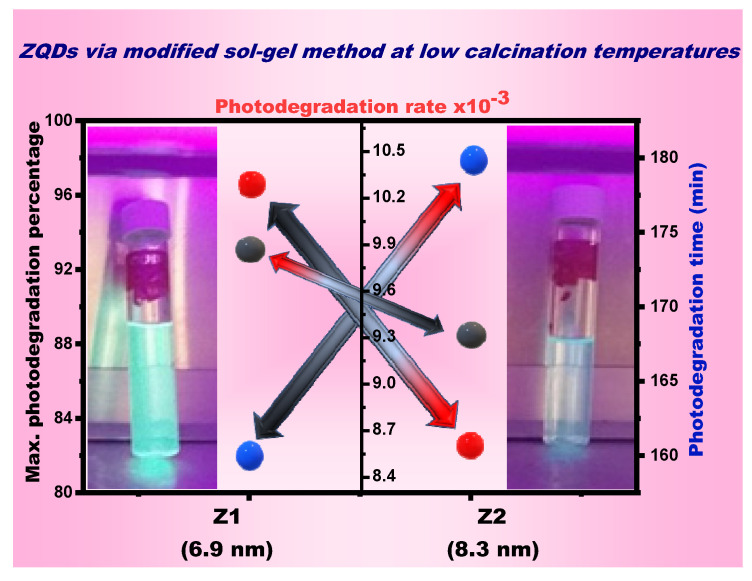
Correlation between Max. photodegradation percentage, photodegradation times, and photodegradation rate of ZQDs.

**Figure 7 nanomaterials-12-02642-f007:**
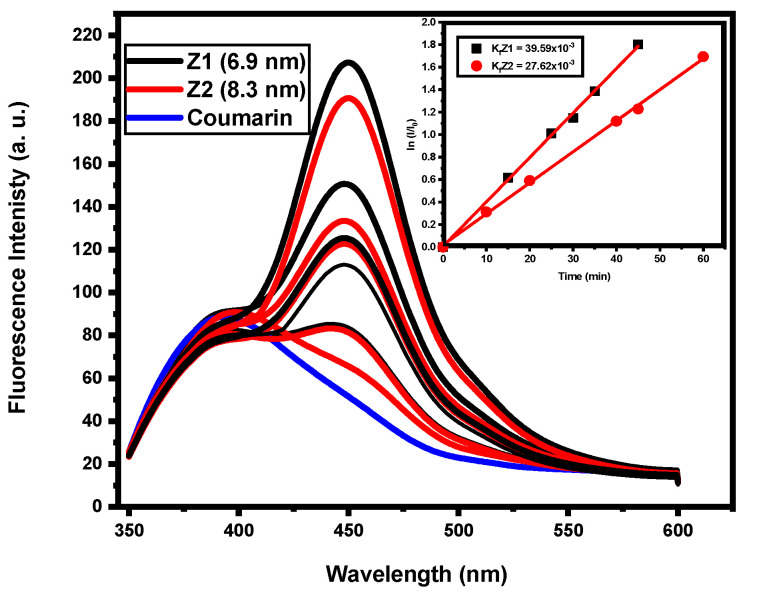
Emission spectra and kinetics plot of coumarin in the presence of ZQDs.

**Figure 8 nanomaterials-12-02642-f008:**
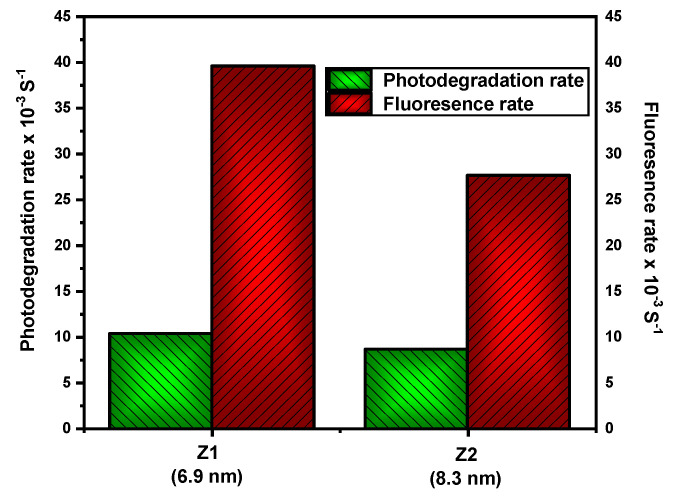
Correlation between photodegradation rate and fluorescence rate for ZQDs.

**Figure 9 nanomaterials-12-02642-f009:**
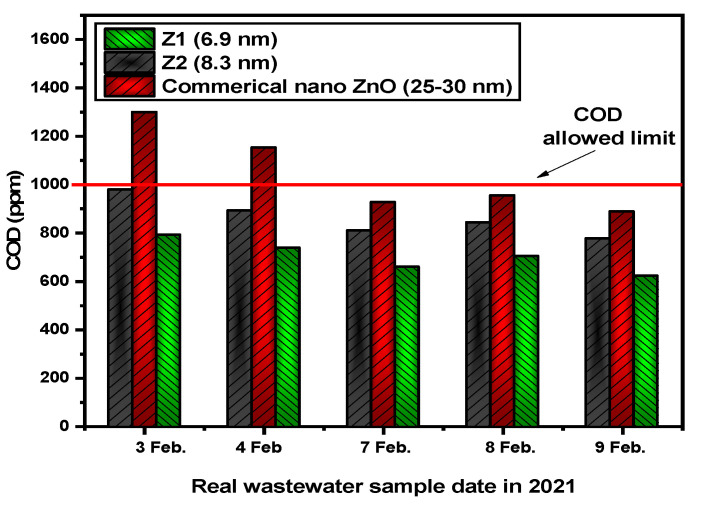
Evaluation of real industrial wastewater during photocatalysis by sunlight via the COD method.

**Figure 10 nanomaterials-12-02642-f010:**
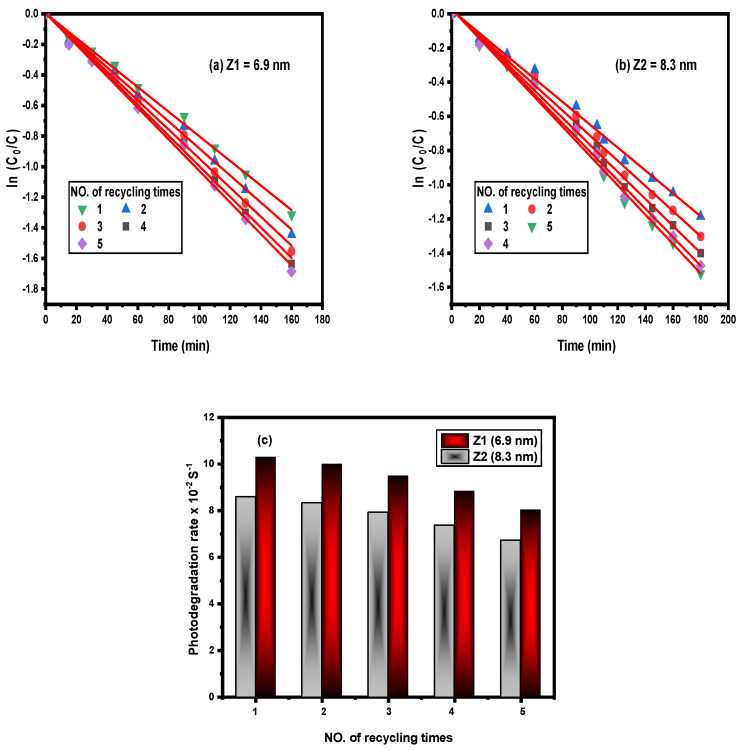
Kinetics plot (**a**,**b**) and correlation between photodegradation rate of reactive yellow dye and no. of recycling processes by using different prepared ZQDs samples (**c**).

**Figure 11 nanomaterials-12-02642-f011:**
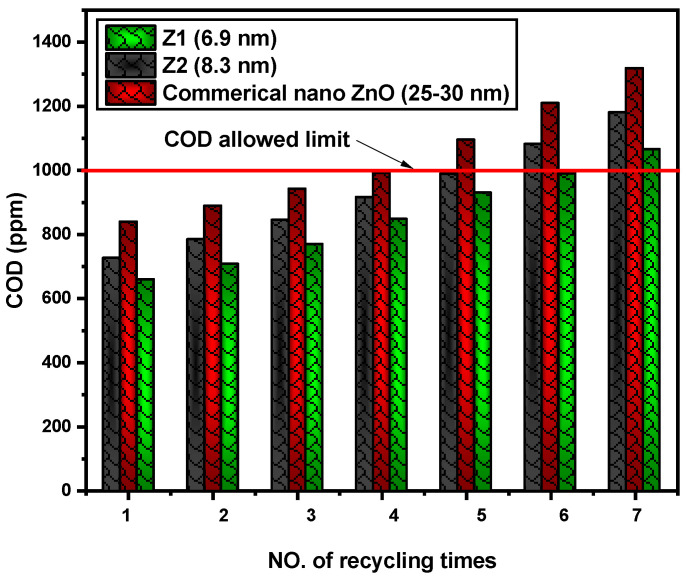
COD limits for the recycling process of real industrial wastewater in the presence of prepared ZQDs samples during photocatalysis by sunlight.

**Table 1 nanomaterials-12-02642-t001:** Fluorescence rate constant (*K_f_*) and photodegradation rate (*K_app_*) for ZQDs prepared samples.

	(Z1 = 6.9 nm)	(Z2 = 8.3 nm)
*K_f_*	39.59 × 10^−3^ S^−1^	27.62 × 10^−3^ S^−1^
*K_app_*	10.30 × 10^−3^ S^−1^	8.59 × 10^−3^ S^−1^

**Table 2 nanomaterials-12-02642-t002:** COD values of real industrial wastewater by the recycling of different prepared ZQDs samples during photocatalysis by sunlight.

Sample Investigated Date	NO. of Recycling Process	COD Value (ppm)		
		Z1 (6.9 nm)	Z2 (8.3 nm)	Nano ZnO (25–30 nm)
3 February 2021	1	658	725	838
4 February 2021	2	706	783	888
7 February 2021	3	768	844	941
8 February 2021	4	847	915	991
9 February 2021	5	929	988	1095
10 February 2021	6	989	1081	1209
11 February 2021	7	1065	1180	1318

Color key: red: Upper limit; yellow: Risky limit; green: Allowed limit.

## Data Availability

All data supporting the conclusions of this article are included within the article.

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
