# Peer review of "Remarkable Recycling Process of ZnO Quantum Dots for Photodegradation of Reactive Yellow Dye and Solar Photocatalytic Treatment Process of Industrial Wastewater"

_nanomaterials, 2022, doi:10.3390/nano12152642_

Round 1
Reviewer 1 Report
In this study, the authors reported the preparation of zinc oxide quantum dots as photocatalyst for the photodegradation of reactive yellow dyes. Sufficient characterization of ZQDs was conducted and the solar photocatalytic treatment of industrial wastewater was performed and the photocatalytic performance was compared with commercial ZnO. However, this work requires major revision before its eventual publication.
(1) What about other methods for preparing ZQDs? Review on the relevant researches is needed.
(2) Did you consider the photolysis of yellow dye by UV? Please provide the blank experiment in the absence of photocatalyst.
(3) There are some minor errors in formats, e.g. line 179 "(110))". Please go through the text carefully and avoid such mistakes.
(4) Fig. 3, When calculating the band gap of ZnO with DRS data, the value of Eg was obtained when Y axis equals to zero.
(5) Section 3.4, The authors mentioned that hydroxyl radicals played key roles in the photocatalysis process. It is suggested that a brief description of how hydroxyl radicals are involved in the photodegradation of reactive yellow dyes can be given in this part. Then, what about the role of other radicals? Please give out evidence if possible.
(6) Section 2.7.1, Line 318-320. It was mentioned that the the photocatalytic performance decreased after 8 recyclings. However, only 5 reuses are found in Figure 10.
(7) In the introduction, it was mentioned that the band gap of ZnO semiconductor material is 3.37. However, the band gaps of the modified Z1 (6.9 nm) and Z2 (8.3 nm) samples are 3.49 eV and 3.44 eV, respectively. Why the photocatalytic reactivity of ZQDs was enhanced while the band gap became wider? What was the key element controlling the photocatalytic performance? Actually, the Eg of prepared samples should be re-calculated according the comments above.
Reviewer 2 Report
Minor revision required
1. In XRD, Page no 4, line no 177-178, it would be better to change the language.
2. The authors should provide the photocatalytic degradation mechanism.
3. High resolution TEM image should be required.
4. Some recent papers related to photocatalytic activities should be cited in proper places
doi.org/10.3390/nano10101960 and doi.org/10.1016/j.jece.2017.10.040
Reviewer 3 Report
The publication by W. Mohamed et al. entitled
“Remarkable recycling process of ZnO quantum dots for photodegradation of Reactive yellow dye and solar photocatalytic treatment process of industrial wastewater" is in a scope of a journal Nanomaterials and is recommended for publishing after performing the suggested modifications.
The work reports on a synthesis of ZnO quantum dots (ZQDs), their characterization by XRD, TEM, BET, determination of the optical properties and testing the synthesized ZQDs for photodegradation of reactive yellow dye, photocatalysis of industrial water and catalyst recycling in yellow dye and industrial water photodegradation. It was shown that the photocatalytic processes are more efficient for ZQDs exhibiting smaller crystallite size and a larger surface area. Better efficiency was shown for the prepared ZQDs catalysis than for commercial ZnO nanoparticles.
The design of the manuscript is appropriate. However, the most important problem noticed is English language, which requires extensive corrections. In majority of sentences a verb is missing. Text is difficult to read and is not clear. Some suggestions of corrections are included in the pdf version of the manuscript and Supplementary Information Figures. The whole text should be corrected according to the suggestions provided in the manuscript. See also comment, which parts should be carefully corrected. Consequently, notation of e.g. Reactive Yellow Dye should be used, i.e. either capital letters or others.
Besides,
(i) Abstract should be corrected, i.e. misprints, style nad grammar. If abbreviation is used full name should be provided before.
It is obvious that the crystallite (nanoparticle) size is inversly proportional to the surface area and therefore catalytic and photocatalytic processes are usualy more efficient for a smaller crystallite size providing a larger surface area. However, the manuscript present the experiemntal results, which may be of an interest for the readers.
(ii) Clearly stated aim of the study should be provided in the last paragraph of section 1. Introduction.
(iii) Section 2.2. Preparation of zinc oxide quantum dots samples is not clearly stated.
(iv) In section 2.3. Characterization, when mentioning a technique applied a producer of an apparatus should be provided.
(v) In section 2.5. Solar photocatalytic activity definition of chemical oxygen demand (COD) is recommended to be provided.
(vi) Section 3.4. lines 247-260 – the process of formation of oxygen radicals and the influence of band gap on photocatalytic activity should be explained more clearly.
(vii) Section 3.5. Photocatalytic activity by fluorescent probe method - provide units for fluorescence rate constant and photodegradation rate in Table 1 and Figure 8.
(viii) Section 2.7.1. Recycling of ZQDs during decontamination of Reactive Yellow dye - provide units in Fig. 10 ( c).
(ix) Section 2.7.1. Recycling of ZQDs during decontamination of Reactive Yellow dye - provide explanation to the statement "replication of images".
(x) Explain a reason for a larger crystallites size obtained during calcination at higher temperature.
(xi) Explain the differences between results obtained by XRD and TEM. In case of TEM no remarkable difference between Z1 and Z2 ZQDs size was obtained (Fig. 2).
(xii) How does the XRD pattern look below 2theta=30 deg? Are any carbon dots visible?

Round 2
Reviewer 1 Report
(1) The authors did not revise the text according to comment (4), I mean the value of Eg was obtained when Y axis equals to zero in Fig. 3, not the XRD pattern in Fig. 1.
(2) Line 345, It should be numbered as 3.7…
(3) Fig. 3, the title of Y-axis should be (αhv)2, not αhv2
